# Isolated Sphenoid Sinus Disease in Children

**DOI:** 10.3390/ijerph20010847

**Published:** 2023-01-02

**Authors:** Michal Kotowski, Jaroslaw Szydlowski

**Affiliations:** Department of Pediatric Otolaryngology, Poznan University of Medical Sciences, 60-572 Poznan, Poland

**Keywords:** sphenoid, isolated sphenoiditis, sphenoid opacification, children

## Abstract

The rarity of isolated sphenoid sinus disease (ISSD) and the specificity of pediatric populations meant that a separate analysis was required in this study. This study aimed to present and discuss the results of an analysis of clinical manifestations, radiological findings and surgical methods based on a large series of exclusively pediatric patients. The study group covered 28 surgically treated children (aged 5.5–17.5 years). The medical data were retrospectively analyzed and meticulously discussed with regards to presenting signs and symptoms, radiographic findings, surgical approaches, complications, post-op care, histopathological results and follow-ups. The dominant symptom was a persistent headache (78%). Four children presented visual symptoms, diplopia in two cases, visual acuity disturbances in one case and both of these symptoms in one patient. Sixteen children presented chronic isolated rhinosinusitis without nasal polyps, six suffered from mucocele and one presented chronic sphenoiditis with sphenochoanal polyp. Four patients turned out to exhibit neoplastic lesions and developmental bony abnormality was diagnosed in one case. No fungal etiology was revealed. The transnasal approach was used in 86% of patients. A transseptal approach with concurrent septoplasty was used in four patients. The patient with visual acuity disturbances completely recovered after the surgery. All children with visual symptoms reported improvement in the vision immediately after surgery. No postoperative complications were observed. Fungal etiology was extremely rare in the pediatric population with ISSD. The surgical treatment should be a minimally invasive procedure regarding a limited range of the pathology in ISSD. Emergency surgery should be performed if ISSD produces any visual loss.

## 1. Introduction

Isolated sphenoid sinus disease (ISSD) is commonly thought of as a rare entity. Usually, it is determined by the presence of the pathology solely in the sphenoid sinus with uni- or bilateral opacification on imaging studies (Figure 1). The term ISSD was introduced by Wyllie et al. in 1973 and was based on a 37-year period of observation at the Mayo Clinic [1]. Since then, an increasing number of cases of sphenoid pathology have been reported, certainly resulting from extremely rapid growth in the availability of CT and MRI [2].

Being a consequence of the common cold or allergic inflammation, paranasal sinus infections are common in children [3]. Sphenoid infections become clinically relevant in children above the age of 5 due to the natural development process of the sinuses [4]. Despite this, ISSD is especially rare in children [5]. According to Gilony et al., the incidence rate among children can be assessed as 0.4% in all paranasal sinuses [6], as compared to 1%–2.7% in the general population with paranasal sinus pathology [1,7,8,9,10,11,12,13,14]. According to Tan and Ong, a lower incidence of isolated sphenoiditis as compared to other sinuses may stem from fewer mucous-secreting cells in the sphenoid epithelium [15].

A spectrum of pathologies may result in the opacification of the sphenoid sinus. Both acute and chronic inflammatory conditions, fungal infections, primary and secondary tumors, or fibrous dysplasia may be the cause [6,16,17,18]. The proximity of the sphenoid sinus to important structures including the pituitary gland, middle fossa dura, optic nerve, cranial nerves (III, IV and VI), pterygoid canals and nerves, pterygopalatine ganglion and artery, internal carotid artery (ICA) and cavernous sinus, occasions the possibility of irreversible neurological consequences or life-threatening complications [1,19,20]. However, these symptoms are usually vague or nonspecific, such as a headache, and sphenoid sinusitis is not usually diagnosed until the patient manifests neurological symptoms [4].

As the symptoms of ISSD in children seem to be more variable and less precisely reported as compared to adults [21], representative analyses of this rare entity are required. Most of the literature covers adults or mixed age groups, and a large series are lacking in the pediatric population [5,14,21,22]. In this study, the authors present one of the largest pediatric cohorts which have been published so far. This study aimed to present and discuss the results of an analysis of clinical manifestations, radiological findings, surgical methods, etiological factors, and histological results based on a large series of exclusively pediatric patients. Moreover, the potential differences in the characteristics of ISSD in children were identified.

## 2. Materials and Methods

The diagnosis of ISSD was based on the most typical signs and symptoms, routine ear, nose and throat (ENT) examinations and radiographic findings. The inclusion criteria were determined as the presence of uni- or bilateral opacification of the sphenoid sinus in radiological studies (computed tomography and/or magnetic resonance imaging) and one of the following: a persistent headache, dizziness or visual symptoms. Patients with the additional presence of pathology in other sinuses were excluded from the study. Twenty-eight children were included in the study group. All of the patients were surgically treated at a single institution between January 2013 and December 2019. The treatment procedure included endoscopic endonasal surgery and perisurgical antibiotic therapy. The extent of the surgery depended on the suspected pathology. Biopsies were taken in each case in order to identify the nature of the pathology. The fungal detection procedure included direct techniques, stained preparations (swabs and histopathological) and cultures. The age of the participants ranged from 5.5 to 17.5 years (mean age 12). The male-to-female ratio was 3:4. Their medical data were retrospectively analyzed with regards to presenting signs and symptoms, radiographic findings, surgical approaches, complications, post-op care, microbiological and histopathological results, and follow-ups. Due to the rarity of the pathology and inclusion criteria, the sample size was considered large and representative. 

The appropriate institutional review board (IRB) approval was obtained for this study (KB-917/21).

## 3. Results

The dominating symptom was a persistent headache, as reported by 22 patients (78%). The duration ranged from 6 weeks to 30 months (mean 12 months). Ten out of twenty children suffered from a localized headache, eight experienced this symptom in more than one region and four were unable to describe the precise location of the headache. One patient made a complaint of dizziness but without a headache. Four children presented visual symptoms, including diplopia and visual acuity disturbances in three cases and one child, respectively. 

Flexible nasoendoscopy was performed in 26 cases. In 93% of children, it was normal. Positive findings were revealed only in two cases: the first patient presented nasal congestion (simultaneously with persistent headaches) and the second presented nasal obstruction due to the presence of sphenochoanal polyp. 

All the participants were investigated radiologically. Diagnosis was based only on the results of CT scans in 20 patients. Contrast enhancement was used in case of suspicion of sphenoid mass. In seven cases, both MRI and CT scans were performed. In one child the diagnosis was made solely on MRI. 

The surgical approach was tailored according to anatomic conditions. The extent of surgery depended on the type and range of sphenoid sinus pathology. A transnasal approach was used in 86% of patients (24/28). The sphenoid was entered through the anterior bony face (13 children) or—if visible—through the surgically enlarged natural ostium (11 cases). No resection of the posteroinferior portion of the middle turbinate was performed in order to facilitate access to the sphenoethmoidal recess. In two cases, septal deviation was corrected to facilitate this transnasal approach.

The transseptal approach with concurrent septoplasty was used in four patients. Patients with suspected neoplasms or lesions with impending intracranial or orbital complications underwent sphenoidotomy with biopsy for histopathological examination. No patients required a transpterygoidal approach. An image-guided intraoperative monitoring tool was not used in any case. No intraoperative complications were reported. 

The type of sphenoid pathology varied in the study group. Sixteen children presented chronic rhinosinusitis without nasal polyps, six suffered from mucocele and one presented chronic sphenoiditis with sphenochoanal polyp. No fungal infections were revealed. Four lesions turned out to be neoplastic lesions (lipoma, osteoma, sphenoid infiltration in the course of acute lymphoblastic leukemia (ALL) and well differentiated chondrosarcoma G1). Fibrous dysplasia was diagnosed in one case.

Post-operative care included the intravenous administration of antibiotic agents. A nasal package was routinely removed 1 day after surgery. No important bleeding was observed. In 26 out of 28 patients the postoperative hospitalization period ranged from 2 to 5 days (a mean of 3 days). No postoperative complications were observed. Systemic steroids were administered in a postoperative period in 21.5% of patients. All the patients perioperatively obtained antibiotics over the following 6 or 7 consecutive days. Topical nasal steroids were prescribed at the hospital discharge and continued for 3 months.

The follow-up period ranged from 24 to 48 months and referred to 24 patients. Patients with neoplastic diseases were referred to the Pediatric Oncology Department. Both children with nasal symptoms reported complete improvement after the surgery. The patient with visual acuity disturbances recovered completely after the surgery. Three children presenting with diplopia reported normalized vision shortly after surgery. Seventeen out of twenty-two children with headaches (77%) reported total relief or a significant reduction in the intensity of symptoms after surgical treatment. Five patients did not improve, and they reported headache in the following months remaining under neurological supervision.

The results of the study are summarized in Table 1. 

## 4. Discussion

Based on the major published series, inflammatory etiologies are responsible for 61–82% of isolated sphenoid lesions [12,14,23,24,25,26]. This figure is similar in the pediatric population, reaching 64% in studies by Elden et al. [5], 79% in the publication by Wang PP et al. [21] and 82% in our group. There were no acute sphenoiditis cases in our material, whereas Elden et al. presented 5 out of 14 patients with acute sphenoiditis and 4 with mucoceles [5]. Marseglia et al. and Caimmi et al. demonstrated homogenous groups of patients with acute sphenoiditis [4,27]. Their studies confirmed previous statements from the Brussels Consensus Meeting [28] about two forms of the disease: severe acute sphenoiditis and non-severe acute sphenoiditis. The first is coupled with fever and headache and is usually associated with neurological symptoms, with swimming and diving as probable predisposing factors. The second is connected with allergic rhinitis and a headache is the dominating symptom.

Similar to Elden et al. [5] and Wang PP et al. [21], there were no patients with fungal etiology among the inflammatory etiologies in our patient cohort. Friedman et al. reported 20% incidence of fungal infections [26], and Wang ZM et al. [14] and Nour et al. [25] reported 15% incidence. Lin et al. showed that even more than 50% of patients developed fungal sphenoiditis, but it could result from climate conditions in Taiwan [20]. Analysis indicates an extremely low incidence of fungal etiology of ISSD in children.

Neoplastic processes were the second most common problem (14.3%) in our study group. This figure is similar to studies by Wang ZM et al. who reported 13% neoplastic lesions in a series of 122 patients [14], while Lin et al. reported—9 cases [20]. Among pediatric patients, Elden at al. revealed 21% of neoplastic cases [5]—the same as Wang PP et al. [21].

The typical symptoms of sinusitis are commonly known but are unhelpful in the case of isolated sphenoid sinus pathology. The problem with isolated sphenoid sinusitis derives from the fact that this entity does not meet the typical criteria of chronic rhinosinusitis presented in EPOS 2020 [29]. The nonspecific symptoms of ISSD often lead to a delay in diagnosis. The most common symptom reported in the literature was a headache, commonly described as nonspecific in location and quality, but refractory to medical treatment. The incidence of this symptom varies from 33% to 81% [1,7,8,10,14,20,25]. Sethi states that retroorbital headaches are typical for ISSD [13], but many researchers disagree. Chronic sphenoiditis is more likely to present with headaches in comparison to acute sphenoiditis [6]. In our study the role of headaches as a dominating symptom was confirmed (78%), corresponding to the data presented in a meta-analysis published by Moss et al. [30]. These results also agree with the studies of Gilony et al., Castelnuovo et al., Marseglia et al. and Haimi-Cohen et al. in pediatric groups [4,6,17,18], whereas Wang PP et al. reported relatively lower rates (42%) in their pediatric group [21]. The mean time of suffering from headaches was 16 days for acute sphenoiditis and 10–16 months in chronic sphenoiditis [21,31,32,33]. In our series, the mean duration of headache was 12 months.

Orbital disturbances make up the second most commonly reported symptom [14,25,34]. Nour et al. reported visual symptoms in 22.5% of patients [25], whereas Chen et al. did so in 47.8% of cases [35]. The spectrum of orbital symptoms covers reduced visual acuity, visual field defects, visual loss and ptosis, proptosis and diplopia as consequences of cranial nerves III, IV and VI affectation [35,36,37]. Dehiscence along the canal of the optic nerve within the sphenoid sinus is present in 8% of the population, and it is therefore more prone to inflammation and compression [38]. According to Lawson and Reino, as well as Martin et al. [12,39], visual disturbance may occur regardless of the nature of sphenoid pathology. In our group, four patients (14.3%) presented visual symptoms, which was slightly higher compared to the data published by Wang PP et al. (10.5%) [21].

There is no agreement among researchers concerning the occurrence of nasal symptoms in ISSD. Ada et al. [34] and Lawson and Reino [12] indicated a very low incidence, i.e., only a few percent of patients, whereas Wang ZM et al. [14] and Friedman et al. [26] reported higher percentages of 31% and 46%, respectively. Wang PP et al. revealed nasal symptoms in 42% of children, which were a consequence of a high percentage of sphenochoanal polyps in their study group (37%) [21]. Nasal symptoms observed among the participants of our study represented inflammatory processes.

An inspection of sphenoid sinus is difficult, and an endoscopic examination of nasal cavities may appear completely normal despite the fact that significant disease [25,32,40] or only subtle findings are present [14]. This corresponds with our results, as only 2 out of 28 patients (7%) presented nasal symptoms (nasal congestion in one child and an obturating mass of sphenochoanal polyp in the another). This result supports Sethi’s opinion that a normal-looking sphenoethmoidal recess on endoscopy does not exclude sphenoid pathology [13]. Nour et al. revealed that 45% of patients exhibited positive findings in the sphenoethmoidal recess [25], whereas Wang PP et al. found this to be the case in 60% of examined pediatric patients [21]. Moreover, Hughes estimated sensitivity and specificity values to be 84% and 92%, respectively [41]. Marseglia et al. state that nasal endoscopy is a well-tolerated and non-invasive technique that may lead to suspicion of sphenoiditis, even in pediatric patients [4]. In the author’s opinion the meaning of endoscopic nasal examination is of less importance in the pediatric population, especially in non-acute cases. On top of that, rigid endoscopy is not well tolerated in lower age groups.

The diagnosis of ISSD is based on radiographic imaging. An increasing number of reported cases of sphenoid pathology can be attributed to the availability of CT and MRI [2]. Currently, plain sinus radiographies are commonly thought to be useless. There is still an ongoing debate in the literature concerning which investigation is the most effective. CT was the most common initial study, as reported by many researchers [5,25] as well as the authors of this study. Marseglia et al. claim that high resolution CT is the supreme method in diagnosing ISSD [4]. On the contrary, Ng et al. stress MRI as the best investigation of choice [42]. McKay-Davies at al. highlight the safety of MRI as a first-line imaging modality among children [32]. Nevertheless, evaluations made with MRI should be introduced in all cases with suspected mass lesions within the sphenoid sinus, as well as intracranial or orbital complications. Nour et al. advise paying special attention to fungal infections because they are occasionally indistinguishable from malignant tumors [25].

There is a consensus that early diagnoses and interventions are crucial in decreasing the probability of permanent neuroophtalmological consequences. In the case of visual loss resulting from ISSD, emergency surgery should be performed. Nevertheless, there are no established and widely accepted indications for sphenoid sinus surgery in the case of ISSD in the pediatric population. In our opinion, various indications presented by Elden et al. are worthy of further discussion but should be supplemented with the clinical situation of ISSD caused by chronic sphenoiditis with persistent symptoms or complications [5]. This is also asserted by other published studies [4,43].

Different surgical techniques make it possible to access the sphenoid sinus [25,44]. The main endoscopic approaches include transethmoidal, transseptal, transnasal and transpterygoidal routes. The specificity of ISSD requires exposure without unnecessary violation of other structures. In the majority of cases the transnasal approach is optimal [34]. Some authors advise a resection of the posteroinferior portion of the middle turbinate in order to facilitate access to the sphenoethmoidal recess if the natural ostium is invisible. As the surgery should be a minimally invasive procedure, we advise a fenestration of the anterior sinus wall to be performed. Friedman et al. used transethmoidal access in ISSD [26], whereas Bolger propagated a transpterygoidal approach [45]. In our opinion, both of these are unjustified. In the case of diagnosing ISSD with the coexistence of nasal septal deviation, a transcollumellar–transseptal technique may be the procedure of choice. This allows for simultaneous sphenoidotomy, septoplasty and direct midline access in order to expose both sphenoid sinuses, especially in the course of sphenoid mucoceles or fungal sinusitis. 

There is a consensus that timely surgical intervention reduces the risk of complications. Complete resolution rates range from 71% to 100% [35,46]. McKay at al. argue that the pediatric population tends to completely recover, despite the presence of major or neurological complications [32]. In our group, four out of five patients presenting visual or neurological symptoms and signs saw improvements. Lin et al. warned that visual disturbances lasting for more than 6 months were regarded as irreversible and connected with the poor prognosis [20]. Some authors recommend performing endoscopic ostial sphenoidotomy as a routine procedure in the diagnostics of patients with intractable headaches [14]. This opinion is controversial but still worth mentioning.

The authors of this study do not advise any postoperative imaging as a routine procedure. Alazawii et al. repeated CT scans in a few weeks postoperatively and observed clear sphenoid sinuses [47]. Olina et al. states that the resolution progress could be monitored and confirmed with repeat CT or MRI scanning [48]. According to Elden et al., in most instances, the sphenoid sinus did not clear for weeks, suggesting that persistent opacification should not be an indication for further surgery, unless symptoms worsen or if other complications are suspected [5]. This is also similar to the opinions of the authors of the study.

## 5. Conclusions

ISSD is a relatively rare entity but is probably underdiagnosed due to unspecific symptoms. Nasal symptoms are rarely detected and have marginal diagnostic value. Surgical treatment should be a minimally invasive procedure given a limited pathology range in ISSD. Fungal etiology is extremely rare in pediatric populations with ISSD. Emergency surgery should be performed if ISSD generates any visual loss. Apart from potentially serious complications, the most isolated sphenoid sinus diseases can be safely endoscopically treated.

## Figures and Tables

**Figure 1 ijerph-20-00847-f001:**
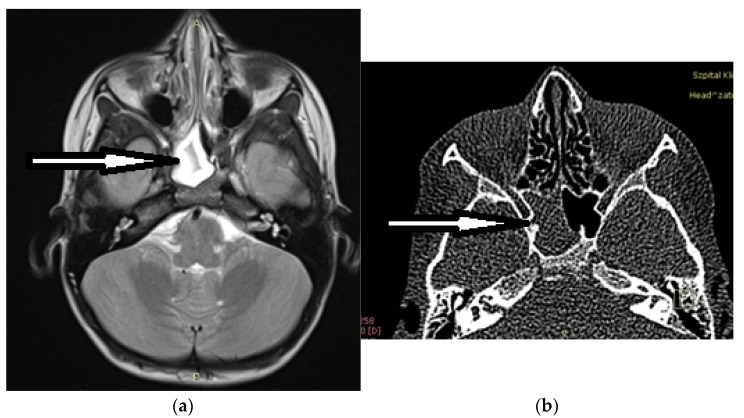
Isolated opacification of the right sphenoid sinus on imaging (arrows): (**a**) MRI T2; (**b**) CT.

**Table 1 ijerph-20-00847-t001:** Characteristics of the study group.

	Sex	Age (Years)	Dominating Symptom	Imaging	Surgical Approach	Pathology	Recovery
1	F	11	Dizziness	CT	TN (no)	Lipoma/L	N/A
2	F	8	Headache	CT	TN (no)	Mucocele/R	Y
3	F	17	Headache	CT	TN (f) + SPL	Osteoma/L	Y
4	M	8.5	Diplopia + Headache	MRI CT	TN (f)	Fibrous dysplasia	except headache
5	F	11	Headache	MRI CT	TN (f)	CRSsNP/L	N
6	F	7.5	Headache	CT	TN (no)	CRSsNP/R	Y
7	M	11	Nasal obstruction	CT	TN (no)	Sphenochoanal polyp/R	Y
8	F	17.5	Headache	CT	TN (no)	CRSsNP/L	Y
9	F	7	Radiological findings	MRI CT	TN (f)	ALL	N/A
10	F	9	Headache	CT	TN (f)	CRSsNP/R	N
11	M	12	Headache	CT	TN (f)	CRSsNP/L	Y
12	F	12.5	Headache	CT	TN (no)	CRSsNP/L	Y
13	F	15.5	Headache + nasal congestion	CT	TS	CRSsNP/L	Y
14	M	17	Headache	CT	TN (f) + SPL	CRSsNP/R	Y
15	M	15	Visual acuity disturbances	MRI	TS	Mucocele/R	Y
16	F	17	Headache	CT	TS	CRSsNP/L	Y
17	M	10	Headache	CT	TN (f)	Mucocele/R	Y
18	F	5.5	Diplopia	MRI CT	TN (no)	CRSsNP/R	Y
19	M	13.5	Headache	CT	TS	CRSsNP/R	N
20	M	5.5	Visual acuity disturbances + Diplopia + headache	MRI CT	TN (f)	CRSsNP/L	except headache
21	M	12.5	Headache	CT	TN (f)	Mucocele/L	Y
22	F	16	Headache	CT	TN (f)	CRSsNP/L	Y
23	F	12	Headache + papilloedema	MRI CT	TN (no)	CRSsNP/L	N
24	M	12	Headache	MRI CT	TN(f)	Mucocele/L	Y
25	F	17	Radiological findings	CT	TN (no)	Well differentiated chondrosarcoma G1	N/A
26	M	14	Headache	CT	TN (no)	CRSsNP/L + R	Y
27	F	14.5	Headache	CT	TN (no)	Mucocele/R	Y
28	M	11.5	Headache	CT	TN (f)	CRSsNP/L	Y

N/A—non-applicable; Sex: M-male, F-female; surgical approach: TN—transnasal, SPL—septoplasty, TS—transseptal, no—natural ostium, f—fenestration; pathology: CRSsNP—chronic isolated sphenoiditis without nasal polyps, ALL—acute lymphoblastic leukemia, R—right side, L—left side.

## Data Availability

The data presented in this study are available on request from the corresponding author. The data is not publicly available due to legal restrictions.

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
