# Peer review of "Isolated Sphenoid Sinus Disease in Children"

_ijerph, 2023, doi:10.3390/ijerph20010847_

Round 1
Reviewer 1 Report
This is a very interesting paper that can be of great value for clinicians. The paper and especially the discussion should gain "readability" by condensing. In the description of the study group I have some concerns
- regarding the group designed as "Chronic sinusitis". What were their problems?
-Vertigo was not mentioned as an initial symptom but appeared in a table. It is well known that younger children seldom report dizziness if not asked. It is often found in adult sphenoiditis. Was it further investigated or only found in this one child?
Reviewer 2 Report
Isolated sphenoid sinus disease in children
International Journal of Environmental Research and Public Health
Isolated sphenoid sinus disease in children
International Journal of Environmental Research and Public Health
Overall this is an interesting paper and is well-written. There are however some minor issues that need to be addressed.
The introduction is short. Since this is considered a rare diagnosis, especially among children it would be of interest to elaborate on the phenomena. Does for instance infections or allergies (for example) paranasal sinus infections, have any influence on ISSD? Even if prevalence is included in the discussion (line 122) should this be included in the introduction?
Line 35, reference is missing.
Line 40, reference is missing.
In lines 42 and 43, the reference is missing.
Line 46-47, states, ….. radiological findings and surgical methods based on the extensive series of exclusively pediatric patients”. Do you consider 28 cases to be a large group of patients?
Suggest making it clear that within a relatively rare diagnosis (and inclusion criteria), 28 cases is a “large” sample.
Line 127, states that this study is “..one of the largest pediatric cohorts which have been published yet”. This information should be presented prior to the discussion.
Figure 1. Please indicate the localization described. I think it should be possible to read this paper for people without medical knowledge.
Reviewer 3 Report
This is a short but meaningful case series on isolated sphenoid sinus disease (ISSD) in the pediatric population. Authors present and discuss the results of an analysis of clinical manifestations, radiological findings and surgical methods based on a series of pediatric patients ( the study group covered 28 surgically treated children). Albeit of interest, the topic seems to be best fit for a medical journal. The use of English language is also far from the required level. Materials and Methods are very poorly described. How was fungal cultures/detection performed? For the above I advise rejection of the paper
